# Comparison of Aerobic Capacity Changes as a Result of a Polarized or Block Training Program among Trained Mountain Bike Cyclists

**DOI:** 10.3390/ijerph18168865

**Published:** 2021-08-23

**Authors:** Paulina Hebisz, Rafał Hebisz, Maja Drelak

**Affiliations:** Department of Physiology and Biochemistry, University School of Physical Education in Wroclaw, 51-612 Wroclaw, Poland; rafalhebisz@poczta.fm (R.H.); maja.drelak.md@gmail.com (M.D.)

**Keywords:** block periodization, polarized training, maximal oxygen uptake

## Abstract

This study compared the effectiveness of a block training program and a polarized training program in developing aerobic capacity in twenty trained mountain bike cyclists. The cyclists were divided into two groups: the block training program group (BT) and the polarized training program group (PT). The experiment lasted 8 weeks. During the experiment, the BT group alternated between 17-day blocks consisting of dominant low-intensity training (LIT) and 11-day blocks consisting of sprint interval training (SIT), and high-intensity interval training (HIIT), while the PT group performed SIT, HIIT, and LIT simultaneously. Before and after the experiment, the cyclists performed incremental tests during which maximal oxygen uptake (VO_2_max), maximal aerobic power (Pmax), power achieved at the first ventilatory threshold (P_VT1_), and at the second ventilatory threshold (P_VT2_) were measured. VO_2_max increased in BT group (from 3.75 ± 0.67 to 4.00 ± 0.75 L∙min^−1^) and PT group (from 3.66 ± 0.73 to 4.20 ± 0.89 L∙min^−1^). In addition, Pmax, P_VT1_, and P_VT2_ increased in both groups to a similar extent. In conclusion, the polarized training program was more effective in developing the VO_2_max compared to the block program. In terms of developing other parameters characterizing the cyclists’ aerobic capacity, the block and polarized program induced similar results.

## 1. Introduction

Competing in mountain bike cycling requires high aerobic capacity [1]. There are several types of training used to develop the aerobic capacity: (1) constant and moderate to low intensity training (LIT) [2,3,4,5]; (2) threshold training (TT), in which efforts are at power levels close to lactate or ventilatory threshold [2,3,6]; (3) high-intensity interval training (HIIT), in which a high-intensity effort is repeated with power close to maximal aerobic power [7,8] or higher than maximal aerobic power [9]; and (4) sprint interval training (SIT), which is performed at maximal intensity (all-out efforts) [10] or close to maximal [11].

Numerous publications have described the effects of using isolated types of training [12,13,14]. Nevertheless, many different types of training influence an athlete’s development in the process of sports training. Matveyev’s periodization model, which describes the distribution of different types of training over an annual training cycle [15], is a popular concept for sports training planning. According to this concept, the initial phase of an endurance athlete’s preparation for the competitive season is dominated by LIT-type training. As the race season approaches, the number and/or volume of LIT-type training decreases, while the volume and number of HIIT-type training increases [16,17]. The effects of simultaneous use of different training types (LIT and HIIT) among endurance athletes were studied by Stöggl and Sperlich [18], calling their concept polarized training. The results of their study demonstrated that a polarized training program allowed for greater development of maximal oxygen uptake (VO_2_max), maximal aerobic power (Pmax), and power at lactate concentrations of 4 mmol∙L^−1^ after a period of 9 weeks, compared to performing one type of training selected from HIIT, TT, or LIT. In the group performing polarized training, VO_2_max increased by 11.7% [18]. In several articles [19,20,21], we described the effects of modifying a polarized training program among cyclists. Our modification consisted of simultaneous training, such as SIT (a dozen or so maximal sprints lasting 30 s each, with rest periods lasting 90 s), HIIT (several efforts lasting 3–4 min with power close to maximal aerobic power), and LIT (training at an intensity of about 80% power measured at the second ventilatory threshold–VT2, taking 2–3 h). Using this training concept yielded even better results than the concept used in the study by Stöggl and Sperlich [18], as VO_2_max increased by 14% after 8 weeks of training [19] and remained at this level for another 4 months of the continuous polarized training program [22]. 

Issurin [23] criticized the concepts of combining multiple training types in a single training cycle, claiming that this approach prevents the achievement of several peaks of performance over the course of a competition season and promotes the development of excessive stress in the body. In the same publication, Issurin [23] presented the theoretical foundations of the block training concept, which were further developed by him in the next years [24,25]. The concept of block training assumes that in a training cycle of 2–4 weeks, the types of training are focused on shaping a minimum number of properties related to the athlete’s qualities for sports competitions; subsequent qualities are shaped in consecutive 2–4-week blocks. However, these publications did not demonstrate the results of research work among endurance athletes [23,24,25]. A detailed block training program was presented by Solli et al. [16] and Rønnestad et al. [26,27,28]. These studies involved alternating blocks of about two weeks of LIT training and blocks of several days or about two weeks of high-intensity training (HIIT and/or SIT). Solli et al. [16] found that the block program was as effective in preparing female cross-country skiing athletes for achieving high sports performance at an elite level as Matveyev’s classical concept. Rønnestad et al. [27] demonstrated an 8.8% increase in VO_2_max after 3 months of performing a block training program. In addition, in one case, VO_2_max was observed to increase from 73.8 mL∙min^−1^∙kg^−1^ to 87 mL∙min^−1^∙kg^−1^ after 58 weeks of such training [28]. 

Having considered the foregoing, the purpose of the presented study was to compare the effectiveness of a block training program and a polarized training program in the development of aerobic capacity in cyclists. We hypothesized that a block training program would contribute more to the development of maximal oxygen uptake, power output at ventilatory thresholds, and maximal aerobic power compared to a polarized training program.

## 2. Materials and Methods

### 2.1. Participants

Twenty national-level mountain bike cyclists participated in the study. The participants were randomly divided into two groups: a group of cyclists performing a block training program (BT group) and a group of cyclists performing a polarized training program (PT group). Each group consisted of 10 cyclists (including 3 women and 7 men, each). The characteristics of the groups are shown in Table 1.

Each participant declared that during the three years prior to entering the experiment, they trained at least 10 h per week (excluding rest periods) and participated in a minimum of 15 cycling races per year.

The study design was approved by the Ethics Committee of the University School of Physical Education and carried out in accordance with the Declaration of Helsinki. Written informed consent was obtained from the participants or their guardians after the study details, procedures, and benefits, and risks were explained.

### 2.2. Course of the Experiment

Prior to the experiment, each participant limited their weekly training volume to 4 sessions (taking approximately 60 min each) for 6 weeks. During this period, experiment participants did not exercise at an intensity exceeding 70% of their maximal heart rate (HRmax). The experiment was conducted over the course of 8 weeks. During the experiment, athletes performed the following types of training:-Sprint interval training (SIT), consisting of 8–12 repetitions at maximal intensity (all-out effort) lasting 30 s. The training was divided into sets (2–3 sets), with 4 maximal repetitions performed in each set. An active rest of 90 s was used between repetitions, during which the participants exercised at a power not exceeding 50 W. Between the sets, an active rest of 15 min was used, during which the exercise was performed at the power reached at the first ventilatory threshold (P_VT1_) (measured during the incremental test–described below). During the first part of the experiment (1st–4th week), all cyclists performed 8 repetitions during SIT. In the following weeks of the experiment (5th–8th week), the cyclists performed 12 repetitions during SIT.-High-intensity interval training (HIIT), consisting of 3–5 efforts lasting 4 min and performed at an intensity of 90% of maximal aerobic power (Pmax) (as measured during the incremental test). An active rest of 6 min was used between these efforts, during which an effort was performed at P_VT1_ intensity. During the first part of the experiment (1st–4th week), all cyclists performed 3 efforts during HIIT. In the following weeks of the experiment (5th–8th week), the cyclists performed 5 efforts during HIIT.-Low-intensity training (LIT) of 2–3 h, performed at P_VT1_ intensity. During the first part of the experiment (1st–4th week), all cyclists performed 2 h of LIT. In the following weeks of the experiment (5th–8th week), the cyclists performed 3 h of LIT.-Passive rest (PR) days were used between the training sessions.

The training program for the BT group was divided into two types of blocks: low-intensity blocks (B_LI_), lasting 17 days, in which LIT-type training predominated, and high-intensity blocks (B_HI_), lasting 11 days, in which only SIT and HIIT training were performed. During the experiment, the foregoing blocks were used alternately, as shown in Figure 1.

The training program for the PT group consisted of 4 training cycles that lasted 14 days each (Figure 2).

During the experiment, cyclists in each group completed 40 training sessions each, with 16 days devoted to passive rest. The BT group completed 11 SIT training sessions, 11 HIIT training sessions, and 18 LIT training sessions, while the PT group completed 12 SIT training sessions, 12 HIIT training sessions, and 16 LIT training sessions.

### 2.3. Laboratory Tests

The participants completed an incremental test before the experiment and 2 days after the experiment. On the following day, after the incremental test, each participant performed a verification test. The tests were performed in controlled laboratory conditions (temperature and humidity controlled) at an Exercise Laboratory of the University School of Physical Education (PN-EN ISO 9001:2001 certified). 

#### 2.3.1. Incremental Test

The test was performed on a Lode Excalibur Sport cycle-ergometer (Lode BV, Groningen, The Netherlands). During the test, exercise started at a load of 40 W in women and 50 W in men, and every 3 min, the load was increased by 40 or 50 W (for women and men, respectively) until the participant refused to continue. The cycle-ergometer was controlled by a computer, which recorded instantaneous power and exercise time. If the participant failed to perform the exercise for 3 min during the last test load, then 0.22 or 0.28 W (for women and men, respectively) was subtracted from the current power value for each missing second. Thus, the value of maximal aerobic power (Pmax) was obtained. 

Respiratory parameters were recorded during the test. The studied cyclists wore a mask connected to a Quark gas analyzer (Cosmed, Milan, Italy), which was calibrated before the test using a reference gas mixture containing 5% carbon dioxide, 16% oxygen, and 79% nitrogen. Respiratory parameters were measured in each recorded breath (breath-by-breath) and then averaged over 30-s intervals. Oxygen uptake (VO_2_), carbon dioxide excretion (VCO_2_), minute pulmonary ventilation (VE), end-tidal partial pressure for oxygen (PETO_2_), and end-tidal partial pressure for carbon dioxide (PETCO_2_) were measured. The highest VO_2_, VCO_2_, and VE recorded during the incremental test or during the verification test (described below) was considered to be maximal oxygen uptake (VO_2_max), maximal carbon dioxide excretion (VCO_2_max), and maximal minute pulmonary ventilation (VEmax), respectively. The first ventilatory threshold (VT1), at a point before the increase in VE∙VO_2_^−1^ and PETO_2_ without a concomitant increase in VE∙VCO_2_^−1^ were determined from the respiratory data recording. The second ventilatory threshold (VT2) was determined at the point before the increase in PETCO_2_, VE∙VCO_2_^−1^, VE∙VO_2_^−1^, according to the methodology described by Pallares et al. [29]. During testing, heart rate (HR) was recorded using a V800 cardiofrequencimeter (Polar, Oy, Finland). Heart rate (HR_VT1_ and HR_VT2_), oxygen uptake (VO_2-VT1_ and VO_2-VT2_), and power (P_VT1_ and P_VT2_) values were indicated when the threshold of VT1 and VT2 occurred. The same procedure as in determining Pmax was used when indicating P_VT1_ and P_VT2_.

#### 2.3.2. Verification Test

A verification test was performed the next day (after the incremental test), also using a Lode Excalibur Sport cycle-ergometer (Lode BV, Groningen, The Netherlands). The test was preceded by a 15-min warm-up, during which the first 5 min was performed at the power achieved at VT1, followed by 10 min of exercise at the power measured midway between the VT1 and VT2 thresholds. After the warm-up, each participant continued to pedal with a 20 W load for 5 min. Verification exercise was then performed at an intensity of 110% Pmax, lasting until refusal. 

During the test, respiratory parameters were measured using the same Quark gas analyzer (Cosmed, Milan, Italy). Measurement of VO_2_, VCO_2_, and VE was used to verify the maximal values of these indices.

During the test, heart rate was also recorded using a V800 cardiofrequencimeter (Polar, Oy, Kempele, Finland). HR measurement was used to verify maximal heart rate.

### 2.4. Statistical Analysis

Before the experiment, the minimum size of the research sample was estimated using the G*Power 3.1.9.4 software. It was assumed that the expected effect size level would be above 0.30, and the expected power of the analysis would be at least 80%. On this basis, the minimum total sample size is 18 people was established.

Statistica 13.1 was used to process the data statistically. The Shapiro–Wilk test was performed in order to assess the normality of the distribution of the examined parameters. Arithmetic mean, standard deviation, and confidence interval were calculated. Analysis of variance with repeated measurements and the Scheffe post-hoc test were used to determine whether there were statistically significant differences in the evaluated parameters between groups and tests.

It was assumed that in each of the performed analyses, a probability level of *p* < 0.05 is required for the result to be statistically significant. 

## 3. Results

Analysis of variance demonstrated a statistically significant main effect of repeated measurements for VO_2_max measured in L∙min^−1^ (*p* = 0.000; F = 81.03; η^2^ = 0.82), VO_2_max measured in mL∙min^−1^∙kg^−1^ (*p* = 0.000; F = 78.31; η^2^ = 0.81), VCO_2_max (*p* = 0.000; F = 21.88; η^2^ = 0.55), VEmax (*p* = 0.003; F = 11.73; η^2^ = 0.39) and Pmax (*p* = 0.000; F = 41.29; η^2^ = 0.70). A statistically significant interaction effect of group and repeated measurements was demonstrated for VO_2_max measured in L∙min^−1^ (*p* = 0.003; F = 11.93; η^2^ = 0.40) and measured in mL∙min^−1^∙kg^−1^ (*p* = 0.003; F = 11.79; η^2^ = 0.40), as well as VCO_2_max (*p* = 0.021; F = 6.37; η^2^ = 0.26). Based on post-hoc tests, it was shown that VO_2_max and Pmax increased statistically significantly in both BT and PT groups. In addition, VCO_2_max and VEmax increased significantly in the PT group (Table 2). 

Analysis of variance demonstrated a statistically significant main effect of repeated measurements for P_VT1_ (*p* = 0.000; F = 24.48; η^2^ = 0.58), VO_2-VT1_ measured in L∙min^−1^ (*p* = 0.000; F = 24.25; η^2^ = 0.57), P_VT2_ (*p* = 0.000; F = 52.07; η^2^ = 0.74), VO_2-VT2_ measured in L∙min^−1^ (*p* = 0.000; F = 66.49; η^2^ = 0.79). A statistically significant interaction effect of group and repeated measurements was demonstrated for VO_2-VT2_ measured in %VO_2_max (*p* = 0.036; F = 5.13; η^2^ = 0.22). Post-hoc tests showed that P_VT1_, P_VT2_ and VO_2-VT2_ increased statistically significantly in both BT and PT groups. In addition, VO_2-VT1_ significantly increased in the PT group (Table 3).

The main effects analysis showed no statistically significant differences in any of the analyzed parameters between the groups.

## 4. Discussion

The presented study demonstrated that both block training and polarized training are effective strategies for developing aerobic capacity among trained cyclists. Both training programs allowed for increases in VO_2_max, Pmax, and power achieved at ventilatory thresholds. However, analysis of variance for VO_2_max showed a statistically significant interaction effect of group and repeated measures, as the increase in this parameter was greater following the polarized training program compared to the block training program. In an earlier study [19], we showed that VO_2_max increased by as much as 14% as a result of 8 weeks polarized training program performed by a group of competitive mountain bike cyclists. We have also shown that a polarized training program leads to an increase in VEmax, the estimated heart stroke volume [19], and endothelial growth factor levels [30]. Based on the papers cited above, it can be concluded that a polarized training program is a stimulus that comprehensively develops the determinants of VO_2_max. The studies presented in this paper confirmed that a polarized training program leads to significant changes in VO_2_max, which increased by 540 mL or 14.8% among cyclists with a baseline VO_2_max of 57.2 mL∙min^−1^∙kg^−1^. Among other authors, Stöggl and Sperlich [18] obtained results similar to those presented in this paper. They showed that after 9 weeks of performing a polarized training program, consisting of LIT and HIIT training (4 × 4 min at an intensity of 90–95% HRmax), VO_2_max increased by 11.7% in a group of athletes with a baseline VO_2_max of 62.6 mL∙min^−1^∙kg^−1^. In a study by Stöggl and Sperlich [18], HIIT training was performed two times per week. However, these studies did not include the use of SIT-type training, which may have resulted in a slightly smaller increase in VO_2_max compared to our study, in which such training was performed. This suggestion comes from data published by Parmar et al. [31]. They demonstrated that among athletes (characterized by a VO_2_max of approximately 60 mL∙min^−1^∙kg^−1^), the size of the increase in VO_2_max in response to a training program is correlated with the weekly total training duration at intensities exceeding power at VT2 [31]. In the present study, more interval efforts were scheduled to be performed. Therefore, the participants trained longer at intensities exceeding the power achieved at VT2, compared to the study by Stöggl and Sperlich [18]. In contrast to the results presented in this paper, Treff et al. [32] showed no change in VO_2_max among rowers resulting from using a polarized training program consisting of LIT and HIIT training. Nevertheless, the results reported by the foregoing authors may differ from the presented study because of two reasons. The first is the lack of SIT training in the protocol by Treff et al. [32]. The second is the difference in baseline performance level. Treff et al. [32] studied highly trained rowers with a baseline VO_2_max of 68 mL∙min^−1^∙kg^−1^, while our study involved trained cyclists with a baseline VO_2_max of 57–60 mL∙min^−1^∙kg^−1^.

Other authors’ studies on polarized training strategies, including LIT and HIIT, have shown improvements in other parameters characterizing aerobic capacity in trained athletes. Stöggl and Sperlich [18] showed that Pmax increased by 5.1%, whereas power at the lactate threshold increased by 8.1%. Similarly, in a study by Neal et al. [33], Pmax and power at lactate threshold increased by 8% and 9%, respectively, among cyclists after a 6 week training period. Furthermore, Muñoz et al. [34] found that using the 10 weeks polarized training program resulted in improved time (by 5%) in a 10-km running competition. In addition, the study presented here showed a significant improvement in Pmax (by 20.4 W or 6.1%) and power at VT1 (by 26.5 W or 16.4%) and VT2 threshold (by 27.6 W or 11.8%), proving the effectiveness of using a training program consisting of LIT, HIIT, and SIT.

In the present study, the increase in VO_2_max resulting from the block training program was smaller compared to the increase achieved as a result of the polarized training program and amounted to 6.7%. The results achieved by other authors who studied the effectiveness of block training programs in the development of VO_2_max are mixed [26,27,35,36]. Breil et al. [35] reported changes in this parameter at 6% in alpine skiers after 11 days of training, involving completing three training blocks consisting of HIIT training (4 × 4 min at an intensity of 90–95% HRmax). Rønnestad et al. [26] achieved a 4.6% increase in VO_2_max in a group of cyclists (with a baseline VO_2_max of 62 mL∙min^−1^∙kg^−1^) after completing a one-week block containing five HIIT-type training and a 3-week block consisting of dominant LIT-type training (over that time, HIIT training was performed once per week). Rønnestad et al. [27] also described the effects of repeating the foregoing training program three times over a 3-month period. They achieved a better effect by doing so, as their VO_2_max increased by 8.8%. On the other hand, VO_2_max did not increase among young cross-country skiers (characterized by a baseline VO_2_max of 60 mL∙min^−1^∙kg^−1^) as a result of a block training program [36]. This program was performed for 6 weeks. During that time, participants alternated between two blocks of HIIT training (with nine training sessions performed in a single block) and two blocks of dominant LIT training. Regardless of differences in the design of block training programs, even the most effective block training program, described by Rønnestad et al. [27], was less effective in developing VO_2_max than the polarized program described in the present study. The polarized training program proposed by us is effective in the development of maximal oxygen uptake because SIT, HIIT, and LIT training can simultaneously stimulate several adaptive mechanisms [19]. These mechanisms include the development of cardiac output by increasing the blood volume, increasing the minute pulmonary ventilation, developing capillaries in the muscles, and intensifying erythropoiesis through efforts to provoke oxygen deficiency [19]. We believe that the above factors may have a cumulative effect in the form of significant changes in VO_2_max. In the block training program, the mechanism described above may have been weaker because the high-intensity training sessions were performed in separate blocks than the low-intensity (and high-volume) training sessions. Perhaps that is why there was no accumulation of the above-mentioned adaptation factors as a result of BT.

One of the limitations of the presented study was the small number of participants (20 cyclists). However, when conducting research among athletes, it is difficult to find a large group of participants representing the same sports discipline, being of similar age, and representing a similar level of performance and aerobic capacity. In addition, some athletes do not consent to participate in scientific research. Another limitation of the presented study was the lack of a group of untrained people. Therefore, it is not known what changes can be expected as a result of the use of polarized or block training programs among untrained people. Simultaneously, it is an indication for future research, checking the effectiveness of the proposed training programs among people with a lower level of aerobic capacity. The training program described in the presented manuscript included low-intensity training and two types of interval training: SIT and HIIT, which resulted in a high exercise load among the studied cyclists. It is not known if such a combination of training would not be too exhausting among untrained people. Moreover, apart from the maximal oxygen uptake, changes in other parameters were similar in the BT and PT groups. As such, further research into the effects of block and polarized training programs may be needed. Coordination of the response of several human body systems to the effort has recently been described as a measure of aerobic capacity [37,38]. Some authors believe that cardiovascular and respiratory systems coordination is a sensitive measure of training effects [39]. Therefore, future research should evaluate changes in cardiovascular and respiratory systems coordination as a result of PT and BT programs.

The presented results can certainly be used to plan training programs by coaches and endurance athletes whose competition is characterized by variable intensity, repeatedly reaching the maximal intensity.

## 5. Conclusions

When applied to a group of trained mountain bike cyclists, the polarized training program proved to be a more effective method to develop VO_2_max than the block training program. In terms of the development of maximal aerobic power and power achieved at ventilatory thresholds, the block and polarized training program induced similar results.

## Figures and Tables

**Figure 1 ijerph-18-08865-f001:**
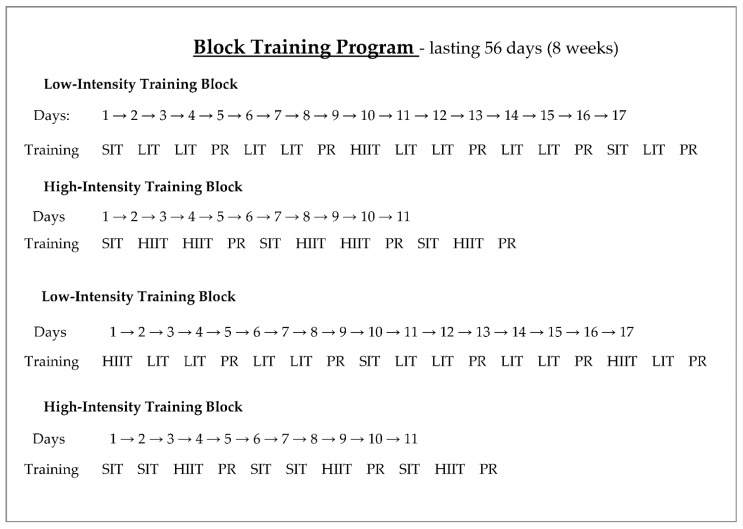
Scheme of the block training program (SIT–sprint interval training; LIT–low-intensity training; PR–passive rest; HIIT–high-intensity interval training).

**Figure 2 ijerph-18-08865-f002:**
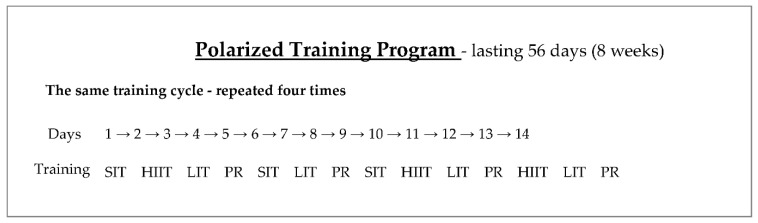
Scheme of the polarized training program (SIT–sprint interval training; HIIT–high-intensity interval training; LIT–low-intensity training; PR–passive rest).

**Table 1 ijerph-18-08865-t001:** Characteristics of experiment participants.

Group	Body Fat (%)	Body Mass (kg)	Body Height (m)	Age (Years)	VO_2_max (mL∙min^−1^∙kg^−1^)	Pmax (W)
BT	12.1 ± 5.3	62.3 ± 8.5	1.73 ± 0.08	18.4 ± 1.6	60.0 ± 4.8	325.5 ± 56.3
PT	10.9 ± 4.9	63.7 ± 8.6	1.74 ± 0.07	18.5 ± 1.9	57.2 ± 5.8	333.2 ± 78.5

VO_2_max—maximal oxygen uptake; Pmax—maximal aerobic power; BT—block training group of cyclists; PT—polarized training group of cyclists; data are presented as mean ± standard deviation.

**Table 2 ijerph-18-08865-t002:** Maximal values of respiratory parameters, heart rate, and aerobic power before and after the experiment in the block training group and the polarized training group.

	Pre-Experiment			Post-Experiment		
	Mean ± SD	95% CI Lower Upper		Mean ± SD	95% CI Lower Upper	
**Block training group of cyclists**						
VO_2_max [L∙min^−1^]	3.75 ± 0.67	3.29	4.22	4.00 ± 0.75 **	3.49	4.50
VO_2_max [mL∙min^−1^∙kg^−1^]	60.0 ± 4.8	56.6	63.4	63.6 ± 4.3 *	60.5	66.6
VCO_2_max [L∙min^−1^]	4.25 ± 0.84	3.64	4.85	4.41 ± 0.89	3.77	5.04
VEmax [L∙min^−1^]	141.8 ± 19.3	128.0	155.7	148.7 ± 28.7	128.2	169.2
HRmax [bpm]	195.3 ± 12.2	186.5	204.1	194.5 ± 8.9	188.1	200.9
Pmax [W]	325.5 ± 56.3	285.2	365.8	347.9 ± 61.7 **	303.8	392.0
**Polarized training group of cyclists**						
VO_2_max [L∙min^−1^]	3.66 ± 0.73	3.14	4.18	4.20 ± 0.89 **	3.56	4.84
VO_2_max [mL∙min^−1^∙kg^−1^]	57.2 ± 5.8	53.1	61.3	65.3 ± 7.8 **	59.7	70.8
VCO_2_max [L∙min^−1^]	4.25 ± 0.81	3.67	4.83	4.79 ± 1.01 **	4.07	5.51
VEmax [L∙min^−1^]	144.6 ± 26.4	125.7	163.5	161.2 ± 31.9 *	138.3	184.0
HRmax [bpm]	190.5 ± 5.6	186.5	194.5	190.4 ± 5.5	186.4	194.4
Pmax [W]	333.2 ± 78.5	277.0	389.4	353.6 ± 75.4 **	299.7	407.5

VO_2_max—maximal oxygen uptake; VCO_2_max—maximal carbon dioxide excretion; VEmax—maximal minute pulmonary ventilation; HRmax—maximal heart rate; Pmax—maximal aerobic power; * *p* < 0.05—significant difference between pre- and post-experiment value; ** *p* < 0.01—significant difference between pre- and post-experiment value; SD—standard deviation; CI—confidence intervals.

**Table 3 ijerph-18-08865-t003:** Power, oxygen uptake, and heart rate achieved at ventilatory thresholds before and after the experiment in the block training group and the polarized training group.

	Pre-Experiment			Post-Experiment		
	Mean ± SD	95% CI Lower Upper		Mean ± SD	95% CI Lower Upper	
**Block training group of cyclists**						
P_VT1_ [W]	152.9 ± 32.9	129.4	176.4	178.8 ± 39.7 *	150.4	207.2
VO_2-VT1_ [L∙min^−1^]	2.45 ± 0.38	2.18	2.72	2.64 ± 0.49	2.29	2.98
VO_2-VT1_ [%VO_2_max]	65.7 ± 4.7	62.3	69.0	67.4 ± 7.0	62.3	72.4
HR_VT1_ [bpm]	158.2 ± 12.5	149.3	167.1	158.7 ± 11.9	150.2	167.2
P_VT2_ [W]	228.0 ± 38.7	200.3	255.7	257.2 ± 37.6 **	230.3	284.1
VO_2-VT2_ [L∙min^−1^]	3.07 ± 0.47	2.74	3.41	3.38 ± 0.45 **	3.06	3.71
VO_2-VT2_ [%VO_2_max]	81.2 ± 3.6	78.6	83.8	85.4 ± 5.9	81.1	89.6
HR_VT2_ [bpm]	179.6 ± 11.2	171.6	187.6	179.4 ± 12.8	170.2	188.6
**Polarized training group of cyclists**						
P_VT1_ [W]	162.0 ± 47.9	127.7	196.3	188.5 ± 53.1 *	150.6	226.4
VO_2-VT1_ [L∙min^−1^]	2.53 ± 0.46	2.20	2.86	2.83 ± 0.56 **	2.43	3.22
VO_2-VT1_ [%VO_2_max]	69.7 ± 7.5	64.3	75.0	67.9 ± 4.8	64.5	71.4
HR_VT1_ [bpm]	149.8 ± 15.7	138.6	161.0	150.9 ± 13.2	141.4	160.4
P_VT2_ [W]	234.1 ± 59.9	191.2	277.0	261.7 ± 58.8 **	219.7	303.7
VO_2-VT2_ [L∙min^−1^]	3.08 ± 0.58	2.66	3.50	3.49 ± 0.65 **	3.02	3.95
VO_2-VT2_ [%VO_2_max]	84.4 ± 7.3	79.1	89.6	83.9 ± 5.5	79.9	87.8
HR_VT2_ [bpm]	172.3 ± 10.4	164.9	179.7	172.1 ± 7.6	166.6	177.6

P_VT1_—power achieved at the first ventilatory threshold; VO_2-VT1_—oxygen uptake achieved at the first ventilatory threshold; HR_VT1_—heart rate achieved at the first ventilatory threshold; P_VT2_—power achieved at the second ventilatory threshold; VO_2-VT2_—oxygen uptake achieved at the second ventilatory threshold; HR_VT2_—heart rate achieved at the second ventilatory threshold; * *p* < 0.05—significant difference between pre- and post-experiment value; ** *p* < 0.01—significant difference between pre- and post-experiment value; SD—standard deviation; CI—confidence intervals.

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
