# Peer review of "Comparison of Aerobic Capacity Changes as a Result of a Polarized or Block Training Program among Trained Mountain Bike Cyclists"

_ijerph, 2021, doi:10.3390/ijerph18168865_

Round 1
Reviewer 1 Report
Congratulations, you are presenting a well-conducted study with an interesting question.
I have a few comments for my part:
- In my opinion, it would be necessary not only to compare the two pre and post training programs, but also to compare the training programs with each other. There are significant differences in parameters between the two training programs. Please state the p-values ​​for the individual parameters between the groups. Is there still a significance in the maximum oxygen uptake ??? In my opinion, there are no major differences in the pre between the two training groups
-and post parameters
- Table 1: Please add muscle mass and fat percentage of the groups
- In the headline they speak of "Moderatly Trained Mountain Bike Cyclists". In my opinion, a maximum relative oxygen uptake of 60 does not correspond to "moderately" but to a very high level.
- Please state in the abstract whether there are significant differences between the groups (i.e. p-values ​​and significance levels).
- Discuss to what extent these results can also be transferred to less well trained groups of people
Author Response
Comments and Suggestions for Authors
Congratulations, you are presenting a well-conducted study with an interesting question.
I have a few comments for my part:
- Thank you very much for your review and suggestions.
- In my opinion, it would be necessary not only to compare the two pre and post training programs, but also to compare the training programs with each other. There are significant differences in parameters between the two training programs. Please state the p-values ​​for the individual parameters between the groups. Is there still a significance in the maximum oxygen uptake ??? In my opinion, there are no major differences in the pre between the two training groups and post parameters
- The main effects analysis showed no statistically significant differences in any of the analyzed parameters between the groups. Similarly, the post-hoc analysis showed no statistically significant differences between the groups in the measurements taken before and after the experiment.
In the current version of the manuscript, we have added this information in the Results section.
- Table 1: Please add muscle mass and fat percentage of the groups
- In the presented study, we assessed the body composition using the Futrex analyzer. This analyzer only measures body fat and lean tissue, it does not analyze muscle mass. Therefore, in Table 1, we only added the body fat value.
- In the headline they speak of "Moderatly Trained Mountain Bike Cyclists". In my opinion, a maximum relative oxygen uptake of 60 does not correspond to "moderately" but to a very high level.
- Indeed, the term "moderately trained mountain bike cyclists" is not appropriate for the studied group of cyclists. However, based on the opinion of other Reviewers, we will use the term "trained mountain bike cyclists".
- Please state in the abstract whether there are significant differences between the groups (i.e. p-values ​​and significance levels).
- There were no significant differences between the groups.
- Discuss to what extent these results can also be transferred to less well trained groups of people
- As suggested, in the Discussion section, a paragraph was added referring to the possible use of the presented results by people with a lower level of aerobic capa city.
Reviewer 2 Report
The manuscript is very well written and clearly presents all the items. The theme is relevant and very important for the sport. However, I suggest that the authors complement some information in the text for better understanding.
1. Better describe the sample used, putting, for example, the levels of competitions that cyclists compete in (international, national).
2. "Discussion": add the limitations and practical applicability of the study.
3. "Conclusion": add suggestions for future studies.
Congrats for the work!
Author Response
Comments and Suggestions for Authors
The manuscript is very well written and clearly presents all the items. The theme is relevant and very important for the sport. However, I suggest that the authors complement some information in the text for better understanding.
- Thank you very much for your review and suggestions.
- Better describe the sample used, putting, for example, the levels of competitions that cyclists compete in (international, national).
- As suggested, in the presented manuscript we described the group of participants as "national level cyclists" because most of the cyclists, - 13 competitors, participated in the national competitions, and only 7 in the international competitions.
"Discussion": add the limitations and practical applicability of the study.
- In the Discussion section, the limitations and practical application of the research were added.
"Conclusion": add suggestions for future studies.
- Suggestions for future studies have been added in the Discussion section.
Congrats for the work!
Reviewer 3 Report
Briefly summarized, this manuscript compared the effectiveness of a block training program and a polarized training program in developing aerobic capacity in twenty moderately trained mountain bike cyclists.The main results indicate that the polarized training program might be more effective in developing the VO2max compared to the block program.
The manuscript is well written and it will be of interest for the readers of IJERPH. I hope the authors accept these comments and criticisms in the manner that was intended; that is, as an effort to offer constructive commentary and advice with a view to strengthening the manuscript:
METHODS
- Did the Authors perform any type of power analysis to determine the sample size for this study?
- Why did you select SIT instead of RST? Please explain.
- Please specify the number of sets/reps for each type of training (SIT, HIIT and LIT). In the current version of the manuscript the Authors provide a range (8-12 reps SIT; 3-4 sets HIIT; 2-3 hours LIT). Do this mean that each participant performed different number of sets/reps? This could significantly impact the results.
- Was there any type of volume/intensity increment during the training period?
- Why did you include SIT sessions in the LIT Block?
- Did the Authors perform a normality test?
DISCUSSION
- Regarding the main result of the study “the polarized training program proved to be a more effective method to develop VO2max than theblock training program” ïƒ The Authors should briefly discuss the physiological mechanisms underlying this result. Why is the polarized training more effective?
-A comment for the Authors: as indicated by previous research, Vo2max may not be sensitive enough to detect specific physiological adaptations occurring after different types of training. A feasible explanation may stem from the fact thatitprovide little information on the specific nonlinear dynamic interactions between organic subsystems. Therefore, it would be interesting to evaluate the effectiveness of PT and BT utilizing variables able to quantify how respiratory and cardiovascular systems coordinate during exercise — this analysis would probably shed some light into the differentiated physiological adaptations provoked by PT and BT. The Authors are encouraged to discuss this point in the discussion (future research?). See below some references regarding this topic that might be of interest for the Authors:
- DOI: 10.1007/s00421-019-04160-3
- DOI: 10.3389/fphys.2020.611550
Author Response
Comments and Suggestions for Authors:
Briefly summarized, this manuscript compared the effectiveness of a block training program and a polarized training program in developing aerobic capacity in twenty moderately trained mountain bike cyclists.The main results indicate that the polarized training program might be more effective in developing the VO2max compared to the block program.
The manuscript is well written and it will be of interest for the readers of IJERPH. I hope the authors accept these comments and criticisms in the manner that was intended; that is, as an effort to offer constructive commentary and advice with a view to strengthening the manuscript:
- Thank you very much for your review and suggestions.
METHODS
- Did the Authors perform any type of power analysis to determine the sample size for this study?
- In order to perform power analysis to determine the sample size, we used the G*Power software (Abt et al. 2020) and performed the repeated measures ANOVA analysis. Based on previous research, we assumed that for the VO2max analysis, we predicted an effect size above 0.30 and we assumed that the power of our calculations should be at least 80%. The minimum sample size for this study, with the above assumptions, was 18.
This information was added at the end of the Materials and Methods section.
Abt G, Boreham C, Davison G, Jackson R, Nevill A, Wallace E. Power, precision, and sample size estimation in sport and exercise science research. Journal of Sports Sciences 2020, 38, 17. doi: 10.1080/02640414.2020.1776002
- Why did you select SIT instead of RST? Please explain.
- To our knowledge, repeated sprint training (RST) is a training consisting of efforts that are mainly based on the dominant phosphagen metabolism, and when the time between the efforts does not allow for phosphocreatine restitution - the dominant glycolytic metabolism. In the literature, RST training consists of repeating efforts such as:
- sprint at a distance of approx. 30 meters (Taylor et al. 2016, Ouergui et al. 2020)
- sprint lasting a few seconds (Brechbuhl et al. 2020)
- change of direction repeated sprint training (Taylor et al. 2016).
We used the term sprint interval training (SIT) in previous studies describing training sessions consisting of repeated sprint efforts that lasted 30 seconds and were similar to the Wingate test. During such efforts, the acid-base balance was disturbed (Hebisz et al. 2016) and a high level of oxygen uptake was achieved (Hebisz et al. 2017). Moreover, other authors also use the term SIT in relation to training characterized by a similar protocol to the one presented by us (Hardcastle et al. 2020, Bertschinger et al. 2019). For the above reasons, we want continue to use the term SIT.
Taylor JM, Macpherson TW, McLaren SJ, Spears I, Weston M. Two Weeks of Repeated-Sprint Training in Soccer: To Turn or Not to Turn? Int J Sports Physiol Perform. 2016 Nov;11(8):998-1004. doi: 10.1123/ijspp.2015-0608.
Ouergui I, Messaoudi H, Chtourou H, Wagner MO, Bouassida A, Bouhlel E, Franchini E, Engel FA. Repeated Sprint Training vs. Repeated High-Intensity Technique Training in Adolescent Taekwondo Athletes-A Randomized Controlled Trial. Int J Environ Res Public Health. 2020 Jun 23;17(12):4506. doi: 10.3390/ijerph17124506.
Brechbuhl C, Brocherie F, Willis SJ, Blokker T, Montalvan B, Girard O, Millet GP, Schmitt L. On the Use of the Repeated-Sprint Training in Hypoxia in Tennis. Front Physiol. 2020 Dec 18;11:588821. doi: 10.3389/fphys.2020.588821. PMID: 33424620; PMCID: PMC7793694.
Hebisz R, Hebisz P, Borkowski J, Zatoń M. Differences in Physiological Responses to Interval Training in Cyclists With and Without Interval Training Experience. J Hum Kinet. 2016 Apr 13;50:93-101. doi: 10.1515/hukin-2015-0147. PMID: 28149346; PMCID: PMC5260645.
Hebisz R, Hebisz P, Zatoń M, Michalik K. Peak oxygen uptake in a sprint interval testing protocol vs. maximal oxygen uptake in an incremental testing protocol and their relationship with cross-country mountain biking performance. Appl Physiol Nutr Metab. 2017 Apr;42(4):371-376. doi: 10.1139/apnm-2016-0362. Epub 2016 Dec 12. PMID: 28177737.
Hardcastle SJ, Ray H, Beale L, Hagger MS. Why sprint interval training is inappropriate for a largely sedentary population. Front Psychol. 2014 Dec 23;5:1505. doi: 10.3389/fpsyg.2014.01505. PMID: 25566166; PMCID: PMC4274872.
Bertschinger R, Giboin LS, Gruber M. Six Sessions of Sprint-Interval Training Did Not Improve Endurance and Neuromuscular Performance in Untrained Men. Front Physiol. 2020 Jan 28;10:1578. doi: 10.3389/fphys.2019.01578. PMID: 32116731; PMCID: PMC7025594.
- Please specify the number of sets/reps for each type of training (SIT, HIIT and LIT). In the current version of the manuscript the Authors provide a range (8-12 reps SIT; 3-4 sets HIIT; 2-3 hours LIT). Do this mean that each participant performed different number of sets/reps? This could significantly impact the results.
- During the first part of the experiment (1st - 4th week), all cyclists performed 8 repetitions during SIT training, 3 sets during HIIT training, and 2 hours of LIT training. In the following weeks of the experiment (5th - 8th week), the training sessions were extended and the cyclists performed 12 repetitions during SIT training, 5 sets during HIIT training, and 3 hours of LIT training. Such information was added to the training descriptions in the Materials and Methods section.
- Was there any type of volume/intensity increment during the training period?
- Yes, there was an increase of volume during the training period as described above.
- Why did you include SIT sessions in the LIT Block?
- In our experiment, we included SIT and HIIT trainings in LIT blocks because we wanted our methodology to be similar to the methodology used by other authors. Ronnestad et al. (2014) and Solli et al. (2019) in LIT blocks, once a week (or training cycle), used high-intensity training. In our study, training with high intensity was SIT and HIIT. Therefore, during the entire experiment in LIT blocks, we applied SIT training three times and HIIT training three times.
Rønnestad BR, Hansen J, Ellefsen S. Block periodization of high-intensity aerobic intervals provides superior training effects in trained cyclists. Scand J Med Sci Sports. 2014 Feb;24(1):34-42. doi: 10.1111/j.1600-0838.2012.01485.x. Epub 2012 May 31. PMID: 22646668.
Rønnestad BR, Ellefsen S, Nygaard H, Zacharoff EE, Vikmoen O, Hansen J, Hallén J. Effects of 12 weeks of block periodization on performance and performance indices in well-trained cyclists. Scand J Med Sci Sports. 2014 Apr;24(2):327-35. doi: 10.1111/sms.12016. Epub 2012 Nov 8. PMID: 23134196.
Solli GS, Tønnessen E, Sandbakk Ø. Block vs. Traditional Periodization of HIT: Two Different Paths to Success for the World's Best Cross-Country Skier. Front Physiol. 2019 Apr 5;10:375. doi: 10.3389/fphys.2019.00375. PMID: 31024338; PMCID: PMC6460991.
- Did the Authors perform a normality test?
- Yes, the Shapiro-Wilk test was done.
DISCUSSION
- Regarding the main result of the study “the polarized training program proved to be a more effective method to develop VO2max than theblock training program” àThe Authors should briefly discuss the physiological mechanisms underlying this result. Why is the polarized training more effective?
- In the final part of the Discussion, we explained the likely causes of the increase in VO2max as a result of polarized training.
A comment for the Authors: as indicated by previous research, Vo2max may not be sensitive enough to detect specific physiological adaptations occurring after different types of training. A feasible explanation may stem from the fact thatitprovide little information on the specific nonlinear dynamic interactions between organic subsystems. Therefore, it would be interesting to evaluate the effectiveness of PT and BT utilizing variables able to quantify how respiratory and cardiovascular systems coordinate during exercise — this analysis would probably shed some light into the differentiated physiological adaptations provoked by PT and BT. The Authors are encouraged to discuss this point in the discussion (future research?). See below some references regarding this topic that might be of interest for the Authors:
- Thank you for recommending interesting articles. Coordination of the physiological response to effort is an interesting aspect. When planning further research projects, we will want to use the parameters assessing the coordination of the cardiovascular and respiratory system reactions. At the end of the Discussion section, we included a paragraph on future research, suggesting the use of methods for assessing cardiovascular and respiratory systems coordination in future studies comparing the effects of BT and PT training programs.